# Accuracy and Normalized Accuracy under Length Bias: Analysis, Guidelines, and a Bayesian Alternative

**Koen Oostermeijer** [1]

## Abstract

Multiple-choice benchmarks that rank candidate completions by conditional log-probability suffer from a length bias: because log-probabilities sum over tokens, longer answers tend to be penalized relative to shorter ones in practice. A common mitigation is to normalize scores by completion length, but we show empirically that this heuristic frequently over-corrects, introducing a bias toward longer answers instead. We first analyze these scoring rules, characterizing when standard and length-normalized accuracy are appropriate and how their length biases depend on the distribution of completion lengths. Motivated by this analysis, we introduce *Bayesian accuracy*, a scoring rule that computes the posterior probability of each candidate under an explicit prior over answer length, thereby removing linear length effects. Bayesian accuracy is a drop-in replacement for likelihood-based multiple-choice evaluation, requires no additional forward passes, and consistently exhibits lower empirical length bias than both standard and length-normalized accuracy across benchmarks and few-shot settings.

## 1. Introduction

Large language models (LLMs) have achieved strong performance on a wide range of natural language processing tasks, from factual question answering to multi-step reasoning (Brown et al., 2020). A common way to evaluate such models is via multiple-choice benchmarks: the model is given a prompt together with a small set of candidate completions, which are ranked according to the conditional log-probability the model assigns to each candidate. The candidate with the highest score is taken as the model's answer, and accuracy is computed as the fraction of prompts

for which this highest-scoring candidate is correct. Prominent examples of this likelihood-based multiple-choice evaluation include MMLU (Hendrycks et al., 2020), HellaSwag (Zellers et al., 2019), and WinoGrande (Sakaguchi et al., 2021).

This way of evaluating models has a key advantage: by restricting the space of possible outputs to a fixed set of benchmark-provided completions, it turns open-ended generation into a simple answer-checking problem. The evaluator no longer has to parse or interpret arbitrary text, it only needs to compare scores over a small, known candidate set. This is particularly useful for pretrained base models that have not been instruction-tuned and may not reliably follow templates or formatting instructions in free-form generation settings. Because multiple-choice evaluation only requires the model to assign probabilities to the given candidates and select the most likely one, it sidesteps these formatting issues and yields a simple, scalable metric that is insensitive to output format.

A key property of these benchmarks is that the total log-likelihood of a completion is typically given by the sum of its token-wise log-probabilities. Because this score is additive over tokens, any per-token uncertainty or modeling error is accumulated with length: even when the model "knows" the correct answer, it is likely to still incur a non-zero loss on each token, so longer answers have more opportunities to accumulate negative log-probability. Empirically, as will be shown in this paper, the total log-likelihoods often decrease approximately linearly with completion length, inducing a systematic *length bias* towards shorter candidates.

A widely used heuristic to mitigate this bias is to normalize the total log-likelihood by the length of the completion, for example by dividing by the number of tokens or bytes (Brown et al., 2020; Gao et al., 2024). Length-normalized scores can be interpreted as (the negative of) an average per-token cross-entropy and are monotonically related to perplexity. In practice, however, we find that such normalization frequently *over-corrects*: scores become biased in the opposite direction, favoring longer completions instead. Moreover, these effects are not uniform across benchmarks or task formats, raising two questions: (I) when is it appropriate to use unnormalized versus length-normalized accu-

[1]Aleph Alpha Research. Correspondence to: Koen Oostermeijer <koen.oostermeijer@aleph-alpha-research.com>.

*Proceedings of the 43^rd International Conference on Machine Learning*, Seoul, South Korea. PMLR 306, 2026. Copyright 2026 by the author(s).

racy, and (II) is there an alternative that behaves robustly across tasks?

Beyond length normalization, several alternative scoring rules have been proposed that adjust for unconditional priors or rescale scores, most notably pointwise mutual information (PMI) and asymmetric normalized PMI (ANPMI); we review these metrics and their limitations as baselines in Section 2.2.

In this work, we study length bias in likelihood-based multiple-choice benchmarks and propose a simple Bayesian alternative to both standard and length-normalized accuracy. Our contributions are as follows:

1. We formalize length bias in log-likelihood-based benchmarks by defining it as the average Kendall rank correlation coefficient between candidate lengths and their scores.
2. Using this formalization, we empirically characterize which benchmarks and scoring rules exhibit substantial length bias and how this depends on model and training setup.
3. Motivated by the empirical observation that, for standard subword tokenizers used by contemporary autoregressive LLMs and across a representative range of multiple-choice benchmarks, total log-likelihoods exhibit a dominant approximately affine dependence on completion length, we introduce *Bayesian accuracy*, a scoring rule that incorporates an explicit length prior and removes this first-order length trend, while remaining a drop-in replacement for existing likelihood-based evaluations.
4. Across benchmarks and few-shot settings, Bayesian accuracy yields lower length bias than standard and length-normalized accuracy, without additional forward passes.

## 2. Background

### 2.1. Notation

We evaluate a language model $f_\theta$ on multiple-choice datasets of the form

$$\mathcal{D} = \left\{ \left( x^{(k)}, C^{(k)}, y^{(k)} \right) \right\}_{k=1}^{K}, \tag{1}$$

where $x^{(k)}$ is the prompt, $C^{(k)} = \{c_m^{(k)}\}_{m=1}^{M_k}$ is the set of candidate completions, and $y^{(k)} \in C^{(k)}$ is the ground-truth answer. These completions are of length $n_m^{(k)} \equiv |c_m^{(k)}|$ measured either in bytes or tokens. For a prompt–completion pair $(x, c)$, we write

$$\ell_\theta(c \mid x) \equiv \log P_\theta(c \mid x) \tag{2}$$

for the total conditional log-probability assigned by the model, obtained by summing token-wise log-probabilities

over the completion.

A *scoring function* $S$ transforms these log-likelihoods into scores

$$s(c \mid x) = S\big(\ell_\theta(c \mid x)\big), \tag{3}$$

that are used to determine the model's prediction by taking the completion with the highest score

$$\hat{y}^{(k)} = \arg \max_{c \in C^{(k)}} s(c \mid x^{(k)}). \tag{4}$$

The resulting accuracy for a given $S$ is the fraction of correct predictions

$$\text{Acc}_S(f_\theta; \mathcal{D}) = \frac{1}{K} \sum_{k=1}^{K} \mathbb{I}\big[\hat{y}^{(k)} = y^{(k)}\big]. \tag{5}$$

### 2.2. Existing Metrics

#### 2.2.1. STANDARD (UNNORMALIZED) ACCURACY

The simplest choice is to use the total conditional log-likelihood directly as the score, in which case the scoring function is the identity function:

$$\text{Standard:} \qquad S_{\text{sta}}(\ell_\theta(c \mid x)) = \ell_\theta(c \mid x). \tag{6}$$

In this case, the selected completion is the one the model assigns the highest overall probability under the conditional distribution $P_\theta(\cdot \mid x)$ to, i.e., which completion is (according to the model) most likely to follow from the prompt. Unless otherwise stated, we refer to *standard accuracy* as the accuracy obtained when $S = S_{\text{sta}}$.

#### 2.2.2. LENGTH-NORMALIZED ACCURACY

A common way of attempting to correct for length bias is to normalize the log-likelihoods by a measure of completion length, typically the number of tokens, characters, or bytes (Brown et al., 2020; Gao et al., 2024). Normalizing by the number of tokens has the advantage that it directly corresponds to text units the model processes and therefore to the number of log-probability terms being summed. The resulting score can be interpreted as the negative average token-wise cross-entropy, or equivalently, the negative log perplexity:

$$\text{Token-normalized:} \quad S_{\text{tok}}(\ell_\theta(c \mid x)) = \frac{\ell_\theta(c \mid x)}{n_{\text{tok}}}. \tag{7}$$

A drawback of token-based normalization is that the metric becomes dependent on the tokenizer, which complicates comparisons between models with different tokenization schemes. This can be mitigated by normalizing by the number of bytes (or characters):

$$\text{Byte-normalized:} \quad S_{\text{byte}}(\ell_\theta(c \mid x)) = \frac{\ell_\theta(c \mid x)}{n_{\text{byte}}}. \tag{8}$$

The latter variant is widely used in practice (Gao et al., 2024; Habib et al., 2023).

### 2.2.3. POINTWISE MUTUAL INFORMATION (PMI)

Pointwise mutual information (PMI) (Fano & Hawkins, 1961) adjusts the conditional log-likelihoods by subtracting the unconditional log-likelihoods of the completions:

$$\text{PMI:} \quad S_{\text{PMI}}(\ell_\theta(c \mid x)) = \ell_\theta(c \mid x) - \ell_\theta(c), \quad (9)$$

where $\ell_\theta(c) \equiv \log P_\theta(c)$ is the log-probability of the completion without conditioning on the prompt. For brevity, we slightly abuse notation and omit the explicit dependence of $\ell_\theta(c)$ on its inputs. Intuitively, PMI measures the increase in log-probability that the prompt $x$ provides for completion $c$, and by doing so removes any prior preference the model might assign to a completion regardless of the context.

Although not as widely used as standard or length-normalized accuracy, PMI-based scoring has appeared in several recent studies (Askell et al., 2021; Biderman et al., 2024). A practical limitation is that PMI requires evaluating $\ell_\theta(c)$ for every completion in addition to $\ell_\theta(c \mid x)$, doubling the number of forward passes needed for evaluation. Moreover, by subtracting the unconditional log-probability of a completion, PMI removes global priors over completions, indirectly mitigating some length-related biases without explicitly performing length normalization.

### 2.2.4. ASYMMETRIC NORMALIZED PMI (ANPMI)

Cho et al. (2025) observe that PMI is bounded from above by $-\ell_\theta(c)$. This happens when the prompt completely determines the completion, which causes the conditional log-likelihood to approach zero (see Equation 9). Consequently, different completions have different maximum attainable PMI values depending on their unconditional probabilities, which can distort comparisons between candidates with substantially different priors. Asymmetric normalized PMI (ANPMI) addresses this by normalizing PMI by $-\ell_\theta(c)$:

$$\text{ANPMI:} \quad S_{\text{ANPMI}}(\ell_\theta(c \mid x)) = -\frac{\ell_\theta(c \mid x) - \ell_\theta(c)}{\ell_\theta(c)}. \quad (10)$$

With this normalization, ANPMI is upper-bounded by 1 for all completions, regardless of their prior probability.

In practice, ANPMI inherits the computational overhead of PMI, since it also requires unconditional log-likelihoods $\ell_\theta(c)$ for all completions. In addition, it can be unstable or undefined for completions with extremely high prior probability (i.e., $\ell_\theta(c) \approx 0$) and is poorly defined for single-token completions in models that do not employ an explicit beginning-of-sentence token. For these reasons, both PMI and ANPMI serve as useful baselines, but they are not ideal

as default scoring rules in large-scale benchmark evaluations. Our proposed Bayesian accuracy, in contrast, only requires conditional log-likelihoods and is explicitly designed to control length bias.

## 3. Experiments

### 3.1. Measuring Length Bias

We quantify the length bias induced by a scoring function $S$ on a multiple-choice dataset by measuring the rank correlation between candidate lengths and their scores within each example: Consider a single example with prompt $x$ and candidate set $C = \{c_m\}_{m=1}^M$. For a model $f_\theta$ and scoring function $S$, the score is defined as

$$s_i = S(\ell_\theta(c_i \mid x)). \quad (11)$$

Throughout the rest of this paper, we primarily use byte length, because of its common usage and desirable tokenizer-independence property. However, length bias can be defined analogously by swapping out byte length for token length. We report additional token-length results in the appendix, which result in the same conclusions.

For a single sample, we measure the correlation between the set of scores $\{s_1, \ldots, s_M\}$ and the set of lengths $\{n_1, \ldots, n_M\}$ using Kendall's rank correlation coefficient $\tau$ (Kendall, 1938). For each unordered pair $(i, j)$ with $1 \leq i < j \leq M$, we compare their scores and lengths:

- The pair is *concordant* if $(s_i - s_j)(n_i - n_j) > 0$, i.e., the candidate with the higher score is also longer.
- The pair is *discordant* if $(s_i - s_j)(n_i - n_j) < 0$, i.e., the candidate with the higher score is shorter.
- The pair is *tied in scores* if $s_i = s_j$.
- The pair is *tied in lengths* if $n_i = n_j$.

Let $M_0 = M(M-1)/2$ be the total number of unordered pairs of candidates, and let $M_c$ and $M_d$ be the numbers of concordant and discordant pairs, respectively. The numbers of tied pairs are given by

$$T_s = \sum_{1 \leq i < j \leq M} \mathbb{I}[s_i = s_j], \quad (12)$$

$$T_b = \sum_{1 \leq i < j \leq M} \mathbb{I}[n_i = n_j], \quad (13)$$

where $\mathbb{I}[\cdot]$ is the indicator function. Using these quantities, Kendall's $\tau_b$ is then computed as

$$\tau(S; C) = \frac{M_c - M_d}{\sqrt{(M_0 - T_s)(M_0 - T_b)}}. \quad (14)$$

Intuitively, the numerator captures the net agreement between the score and length orderings: it is positive when pairs where the higher-scoring candidate is longer dominate, and negative when higher scores tend to go to shorter

candidates. The denominator normalizes by the number of pairs whose relative ordering is actually determined once ties in scores and lengths are removed, ensuring that $\tau(S; C) \in [-1, 1]$. Negative values indicate that higher scores are typically assigned to *shorter* candidates, positive values indicate a tendency to favor *longer* candidates, and values near zero correspond to little or no monotonic association between score and length. When all candidates have identical scores or identical lengths, no meaningful ordering exists and the denominator is zero, so $\tau(S; C)$ is undefined.

The overall length bias of a scoring function $S$ on dataset $\mathcal{D}$ is defined as the average Kendall correlation across all samples for which it is defined. Let

$$I_{\text{def}} \equiv \left\{ k \in \{1, \ldots, K\} \mid \tau\left(S; C^{(k)}\right) \text{ is defined} \right\} \quad (15)$$

be the set of indices with a well-defined $\tau$. Then, the overall length bias of the dataset is computed as the average over $\tau$-defined samples:

$$\tau(S, \mathcal{D}) \equiv \frac{1}{|I_{\text{def}}|} \sum_{k \in I_{\text{def}}} \tau(S; C^{(k)}). \quad (16)$$

## 3.2. Benchmarks and Models

We evaluate on a suite of multiple-choice benchmarks: ARC (Clark et al., 2018), ARC German (LAION, 2024), HellaSwag (Zellers et al., 2019), MMLU (Hendrycks et al., 2020), OpenbookQA (Mihaylov et al., 2018), SciQ (Johannes Welbl, 2017), and WinoGrande (Sakaguchi et al., 2021). For MMLU we use two formats. In *MMLU Full-Text*, the prompt contains the question and answer options, and each candidate completion is the full answer text. In *MMLU Cloze*, the option list is removed and the model instead fills in an answer span directly in the context. These variants let us probe how length bias changes when moving from single-symbol labels to heterogeneous multi-token completions. We run all experiments in a zero-shot setting; prompt templates and preprocessing details are given in the appendix.

We evaluate open-weight models from the LLaMA, Qwen, Mistral, Pythia, and SmolLM families, covering parameter counts from 135M to 70B in both base and instruction-tuned configurations (Grattafiori et al., 2024; Jiang et al., 2023; Biderman et al., 2023; Allal et al., 2024; Qwen Team, 2025).

Unless otherwise stated, experiments in the main text use the zero-shot setting; few-shot results are reported in the appendix. In the few-shot setting, we use one fixed exemplar set per prompt configuration, and estimate $\hat{b}$ separately for each model–dataset–prompt configuration, including the corresponding few-shot prompt.

All evaluations are run using the Aleph Alpha Eval Framework (Aleph Alpha Research, 2025).

## 3.3. Empirical Length Bias of Existing Scoring Rules

We begin by analyzing how completion length relates to log-likelihood under the standard and length-normalized scoring rules, and how these score-length curves align with the measured length biases. This lets us check how well the assumed linear scaling of log-likelihood with length holds in practice and how these trends translate into the rank-based length bias $\tau(S, \mathcal{D})$, as defined in Section 3.1.

### 3.3.1. STANDARD ACCURACY

Figure 1 plots, for each benchmark, the average conditional log-likelihood $\ell_\theta(c \mid x)$ as a function of completion byte length $n_{\text{byte}}$, aggregated over all models and candidates and binned for readability.

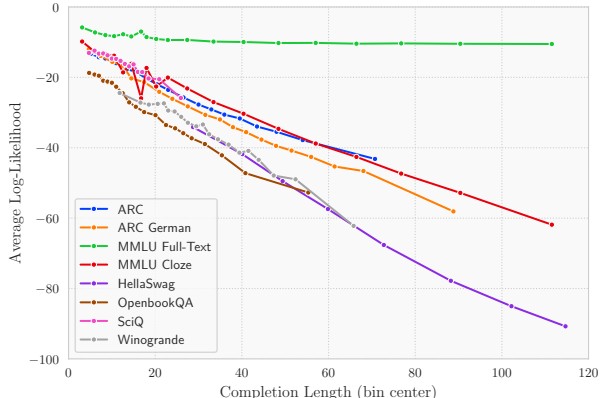

*Figure 1.* Average conditional log-likelihood $\ell_\theta(c \mid x)$ versus completion byte length $n_{\text{byte}}$, aggregated over all models. For readability, completion lengths are restricted to a maximum of 120 bytes and results are binned.

Across almost all benchmarks we observe a pronounced, approximately linear decrease of total log-likelihood with length. The main exception is MMLU Full-Text, whose slope is close to zero at the aggregate level. This behavior matches the intuition that any non-zero per-token loss accumulates across tokens and therefore induces a systematic preference for shorter candidates. It is worth noting that the fitted lines also exhibit a substantial positive intercept at length one. This seems to indicate that the first token of the completion is noticeably harder to predict than subsequent tokens. Initial tokens must both start a new span (often with weaker local context than mid-span tokens). Subsequent tokens then benefit from this additional context and incur a smaller average per-token loss.

Table 1 summarizes the actual length biases for standard accuracy using the average Kendall correlation $\tau(S_{\text{sta}}, \mathcal{D})$. For most benchmarks and models, $\tau$ is clearly negative, confirming our suspicion that unnormalized likelihood tends to favor shorter candidates. The effect is particularly strong

on HellaSwag and the ARC benchmarks, where candidate lengths vary substantially, and weakest on SciQ and Wino-Grande, whose $\tau$ values are close to zero despite visible slopes in Figure 1. This illustrates that a sloped score-versus-length curve does not necessarily imply a strong per-example rank correlation: length bias depends not only on the average slope but also on how candidate lengths are distributed within each question. Within each model family, the magnitude of negative $\tau$ tends to shrink with model size, indicating that larger models are somewhat less sensitive to length artifacts under standard scoring, although the bias remains clearly non-zero.

### 3.3.2. MULTIPLE-CHOICE WITH LOCALLY DETERMINED ANSWERS

MMLU Full-Text provides an illustrative example. Here, each candidate completion is the full answer sentence, but the question and the answer options are already present in the prompt. When we decompose the total log-likelihood into per-token contributions, we find that the *first* one or two answer tokens account for almost all of the variation between candidates. Once the model has committed to a specific option, the remainder of the answer is largely a deterministic copy of text already appearing in the prompt and therefore carries much higher conditional probabilities and much smaller per-token loss.

In other words, for these "locally determined" answers the candidate identity is effectively fixed by the first few tokens, and the remaining tokens contribute an almost identical offset to the log-likelihoods of all candidates. Consequently, the effective slope is close to zero. This explains why MMLU Full-Text shows little aggregate length dependence and near-zero length bias under standard accuracy despite substantial variation in absolute completion lengths.

A second situation in which standard accuracy is effectively insensitive to length is when all completions for a given example have (almost) the same length. Intuitively, if the shortest and longest candidates differ only slightly, then they all sum log-probabilities over nearly the same number of tokens/bytes, so the length-dependent part of the score is almost identical across candidates.

We summarize these observations as a simple rule of thumb:

> **When is standard accuracy safe to use?** Standard (un-normalized) accuracy is mainly appropriate when (i) the answer text already appears in the prompt and the first one or two completion tokens effectively fix the option, or (ii) all candidates for a question have very similar lengths, with single-letter prediction benchmarks as an extreme case.

### 3.3.3. NORMALIZED ACCURACY

We now repeat the analysis for byte-normalized scores $S_{\text{byte}}$, plotting $\ell_\theta(c \mid x)/n_{\text{byte}}$ against $n_{\text{byte}}$ in Figure 2.

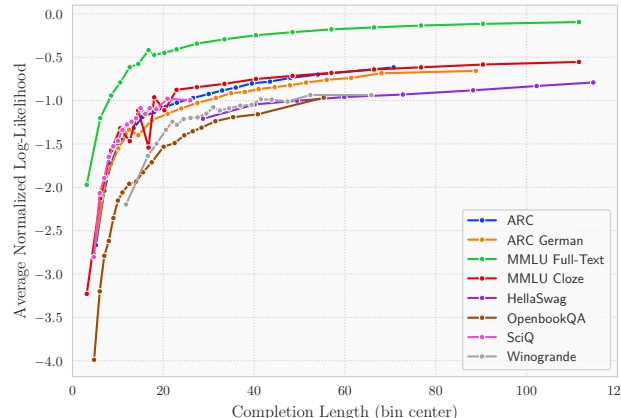

*Figure 2.* Average byte-normalized conditional log-likelihood $\ell_\theta(c \mid x)/n_{\text{byte}}$ versus completion byte length, aggregated over all models. For readability, completion lengths are restricted to a maximum of 120 bytes and results are binned.

Naïve length normalization largely flattens the global trend over medium-length completions: the strong negative slopes in Figure 1 are substantially reduced. However, for short completions we often observe a sharp decrease in $\ell_\theta(c \mid x)/n_{\text{byte}}$ as a function of length. This suggests that normalization can over-correct the raw length bias: very short completions become relatively under-scored compared to slightly longer ones, inducing a systematic bias *towards* longer answers instead.

Table 2 quantifies this effect. In contrast to Table 1, the Kendall correlations $\tau(S_{\text{byte}}, \mathcal{D})$ are uniformly positive across benchmarks and models, often with comparable or even larger magnitude than the negative biases under standard accuracy. On datasets such as ARC, ARC German, OpenbookQA, SciQ and WinoGrande, length-normalization turns a modest negative or near-zero $\tau$ into a substantial positive one, indicating that normalization introduces a strong length bias where there was little before. For HellaSwag and MMLU Full-Text, byte-normalization partially corrects the strong negative bias of standard accuracy, bringing $\tau$ closer to zero for base models, but overshoots for instruction-tuned models, which show markedly larger positive $\tau$ on most benchmarks.

These mixed outcomes make it hard to state a simple, structural condition, analogous to locally determined answers or small within-example length variation for $S_{\text{sta}}$—under which length-normalized accuracy is guaranteed to be safe. Instead, its behavior appears to depend sensitively on the dataset templating and the specific model family. We therefore summarize its practical use in a rule-of-thumb box:

*Table 1.* Length bias for standard accuracy.

| Benchmark | ARC | ARC German | HellaSwag | MMLU Cloze | MMLU Full-Text | OpenbookQA | SciQ | WinoGrande |
|---|---|---|---|---|---|---|---|---|
| Llama 3.2 1B | -0.19 | -0.30 | -0.55 | -0.38 | -0.07 | -0.33 | -0.07 | -0.06 |
| Llama 3.2 3B | -0.17 | -0.26 | -0.52 | -0.38 | 0.06 | -0.27 | -0.05 | 0.01 |
| Llama 3.1 8B | -0.15 | -0.24 | -0.49 | -0.33 | 0.03 | -0.25 | -0.05 | -0.05 |
| Llama 3.1 70B | -0.14 | -0.20 | -0.46 | -0.30 | 0.04 | -0.26 | -0.04 | -0.03 |
| Qwen3 600M | -0.20 | -0.26 | -0.58 | -0.38 | 0.02 | -0.30 | -0.06 | 0.00 |
| Qwen3 1.7B | -0.17 | -0.26 | -0.54 | -0.35 | -0.02 | -0.29 | -0.05 | -0.01 |
| Qwen3 4B | -0.17 | -0.21 | -0.52 | -0.34 | 0.01 | -0.27 | -0.04 | -0.02 |
| Qwen3 8B | -0.15 | -0.20 | -0.50 | -0.35 | 0.04 | -0.27 | -0.04 | -0.02 |
| Qwen3 14B | -0.15 | -0.21 | -0.48 | -0.32 | 0.04 | -0.25 | -0.05 | -0.01 |
| Mistral 7B | -0.16 | -0.25 | -0.49 | -0.35 | 0.03 | -0.29 | -0.06 | -0.07 |
| Pythia 410M | -0.27 | -0.40 | -0.61 | -0.47 | -0.21 | -0.35 | -0.13 | -0.05 |
| SmolLM 135M | -0.22 | -0.44 | -0.60 | -0.43 | -0.16 | -0.30 | -0.09 | -0.04 |
| Qwen3 600M Instruct | -0.18 | -0.26 | -0.58 | -0.37 | 0.02 | -0.25 | -0.07 | 0.00 |
| Qwen3 1.7B Instruct | -0.13 | -0.23 | -0.55 | -0.26 | 0.03 | -0.24 | -0.05 | 0.08 |
| Qwen3 4B Instruct | -0.14 | -0.18 | -0.55 | -0.27 | 0.03 | -0.20 | -0.04 | 0.11 |
| Qwen3 8B Instruct | -0.13 | -0.17 | -0.52 | -0.26 | 0.06 | -0.19 | -0.04 | -0.01 |
| Qwen3 14B Instruct | -0.12 | -0.17 | -0.52 | -0.22 | 0.07 | -0.16 | -0.03 | 0.05 |

*Table 2.* Length bias of byte-normalized accuracy.

| Benchmark | ARC | ARC German | HellaSwag | MMLU Cloze | MMLU Full-Text | OpenbookQA | SciQ | WinoGrande |
|---|---|---|---|---|---|---|---|---|
| Llama 3.2 1B | 0.31 | 0.34 | 0.05 | 0.27 | 0.49 | 0.36 | 0.27 | 0.44 |
| Llama 3.2 3B | 0.26 | 0.29 | 0.04 | 0.27 | 0.48 | 0.36 | 0.23 | 0.42 |
| Llama 3.1 8B | 0.22 | 0.29 | 0.05 | 0.23 | 0.44 | 0.38 | 0.22 | 0.34 |
| Llama 3.1 70B | 0.22 | 0.29 | 0.05 | 0.25 | 0.34 | 0.38 | 0.24 | 0.31 |
| Qwen3 600M | 0.37 | 0.31 | 0.05 | 0.25 | 0.40 | 0.37 | 0.31 | 0.40 |
| Qwen3 1.7B | 0.28 | 0.29 | 0.05 | 0.23 | 0.29 | 0.38 | 0.27 | 0.40 |
| Qwen3 4B | 0.25 | 0.29 | 0.04 | 0.25 | 0.29 | 0.37 | 0.27 | 0.30 |
| Qwen3 8B | 0.23 | 0.27 | 0.04 | 0.23 | 0.26 | 0.36 | 0.26 | 0.30 |
| Qwen3 14B | 0.22 | 0.28 | 0.04 | 0.24 | 0.33 | 0.38 | 0.25 | 0.28 |
| Mistral 7B | 0.23 | 0.28 | 0.04 | 0.24 | 0.40 | 0.40 | 0.22 | 0.33 |
| Pythia 410M | 0.39 | 0.32 | 0.07 | 0.28 | 0.33 | 0.37 | 0.36 | 0.48 |
| SmolLM 135M | 0.33 | 0.27 | 0.06 | 0.25 | 0.42 | 0.38 | 0.29 | 0.41 |
| Qwen3 600M Instruct | 0.43 | 0.38 | 0.33 | 0.34 | 0.24 | 0.60 | 0.44 | 0.47 |
| Qwen3 1.7B Instruct | 0.41 | 0.43 | 0.32 | 0.33 | 0.30 | 0.63 | 0.41 | 0.46 |
| Qwen3 4B Instruct | 0.40 | 0.43 | 0.35 | 0.40 | 0.32 | 0.61 | 0.45 | 0.44 |
| Qwen3 8B Instruct | 0.34 | 0.43 | 0.34 | 0.30 | 0.28 | 0.65 | 0.33 | 0.41 |
| Qwen3 14B Instruct | 0.40 | 0.40 | 0.35 | 0.38 | 0.30 | 0.69 | 0.36 | 0.44 |

**When is length-normalized accuracy safe to use?** The results do not reveal a simple heuristic that guarantees low length bias under byte- or token-normalized accuracy. We therefore do not view normalized accuracy as a safe default and only recommend it once low length bias has been empirically confirmed. Instead, we suggest using the length-corrected Bayesian alternative introduced in the next section.

## 4. Bayesian Accuracy

### 4.1. A Generative Model for Log-Likelihoods

We now formalize the empirical observation from Section 3.3 that, within a benchmark, total conditional log-likelihoods of standard subword tokenizers scale approximately linearly with completion length. For each example $k$ with prompt $x^{(k)}$ and candidate set $C^{(k)} = \{c_m^{(k)}\}_{m=1}^{M_k}$, write $\ell_m^{(k)} \equiv \ell_\theta(c_m^{(k)} \mid x^{(k)})$ for the total conditional log-likelihood, and let $n_m^{(k)}$ denote the byte length of $c_m^{(k)}$.

A simple but flexible description of the observed score-length relationship is the hierarchical linear model

$$\ell_m^{(k)} = \alpha_k + \beta_k n_m^{(k)} + \varepsilon_m^{(k)}, \qquad (17)$$

where

- $\alpha_k$ is a prompt-specific offset capturing the overall difficulty of item $k$ and the effect of the first completion token,
- $\beta_k$ is a prompt-specific average per-byte contribution to the log-likelihood, i.e., the local slope of the length trend for example $k$,
- $\varepsilon_m^{(k)}$ aggregates token-level fluctuations around this trend and has zero mean with a variance that grows linearly in the length.

We treat $(\alpha_k, \beta_k)$ as draws from a population distribution with finite moments $\mathbb{E}[\alpha_k] = a$, $\mathbb{E}[\beta_k] = b$ and variances $\mathrm{Var}[\alpha_k] = \sigma_\alpha^2$, $\mathrm{Var}[\beta_k] = \sigma_\beta^2$. Marginalizing over prompts yields a linear trend:

$$\mathbb{E}[\ell_m^{(k)} \mid n_m^{(k)} = n] = a + bn, \qquad (18)$$

and variance that scales quadratically in the length:

$$\text{Var}[\ell_m^{(k)} \mid n_m^{(k)} = n] = \mathcal{O}(n^2). \qquad (19)$$

This is in line with the empirical trends in Figure 1 and Figure 3 in the Appendix. The quadratic behavior stems from variation in the per-example slopes $\beta_k$: if $\beta_k$ were constant, only the linear $n\sigma^2$ term would remain.

If we use the raw $\ell_m^{(k)}$ to rank candidates, the average slope $b$ induces a systematic preference for shorter completions when $b < 0$ or longer completions when $b > 0$. Our goal is to remove this *global* length trend while preserving genuine, context-dependent preferences among candidates of similar length. Rather than dividing by length, which rescales both signal and noise, we instead introduce an explicit prior over completion lengths and absorb the linear trend into that prior.

## 4.2. Bayesian Correction

Let $P_\theta(c \mid x)$ denote the model's conditional distribution over completions and $\ell_\theta(c \mid x) = \log P_\theta(c \mid x)$ the corresponding log-likelihood. We introduce a prior over completion lengths, $P_{\text{prior}}(n)$, and define a posterior over completions

$$P_{\text{post}}(c \mid x) \propto P_\theta(c \mid x)\, P_{\text{prior}}(n). \qquad (20)$$

Taking the logarithm of the posterior distribution gives

$$\log P_{\text{post}}(c \mid x) = \ell_\theta(c \mid x) + \log P_{\text{prior}}(n) - \log Z(x), \qquad (21)$$

where $Z(x)$ is a normalization constant that does not depend on $c$ and therefore does not affect the argmax over candidates.

The linear relation (18) implies that the length prior is exponential in length:

$$P_{\text{prior}}(n) \propto \exp(-b\,n), \qquad (22)$$

with length slope $b \in \mathbb{R}$. Since benchmark evaluation only compares a finite set of candidates, we only need these relative length weights within each candidate set. Plugging this into the posterior and dropping terms that are constant across candidates yields the *length-debiased log-likelihood score*:

$$S_{\text{Bayes}}(c \mid x; b) \equiv \tilde{\ell}_\theta(c \mid x; b) \equiv \ell_\theta(c \mid x) - b\,n. \qquad (23)$$

Because $\tilde{\ell}_\theta$ is obtained by a simple affine transformation of the same conditional log-likelihoods used for standard accuracy, Bayesian accuracy can be used as a drop-in replacement that does not require any additional model evaluations. It should be noted that this correction targets length-dependent priors induced by additive log-likelihood accumulation, but does not remove other answer priors such as preferences for frequent words, phrases, or syntactic forms.

## 4.3. Estimating the Length Decay Factor

To apply Bayesian accuracy, we estimate one global length-decay factor $b$ for each model–dataset–prompt configuration. Rather than fitting a pooled regression of log-likelihood on length, we estimate $b$ from within-question variation. This removes prompt-specific offsets $\alpha_k$, which would otherwise confound the slope whenever easier or harder prompts systematically contain longer candidates.

Concretely, for each question we center candidate lengths and log-likelihoods within that question and fit the resulting fixed-effect slope:

$$\hat{b} = \frac{\displaystyle\sum_k M_k \sum_{i=1}^{M_k} \big(n_i^{(k)} - \bar{n}^{(k)}\big)\big(\ell_i^{(k)} - \bar{\ell}^{(k)}\big)}{\displaystyle\sum_k M_k \sum_{i=1}^{M_k} \big(n_i^{(k)} - \bar{n}^{(k)}\big)^2}. \qquad (24)$$

This estimator is equivalent to regressing pairwise log-likelihood differences on pairwise length differences, but the centered form is linear in the number of candidates per question. When calibration data contain little within-question length variation, $\hat{b}$ can be noisy; in that case we use the same prompt configuration as the evaluation set and can optionally stabilize the estimate with held-out calibration, pooling across related templates, or shrinkage toward $b = 0$. Appendix A.1 gives the full derivation and Algorithm 1 gives the implementation.

## 4.4. Results

For each model–dataset pair we estimate a separate $\hat{b}$ and rescore candidates with the length-corrected log-likelihood $\tilde{\ell}_\theta(c \mid x; \hat{b})$. Table 3 reports the resulting Kendall correlations $\tau(S, \mathcal{D})$ between candidate length and score under Bayesian accuracy.

Bayesian accuracy brings the rank-correlation between candidate length and score close to zero: almost all entries in Table 3 satisfy $|\tau| \leq 0.1$. This supports Bayesian accuracy as a robust scoring rule that substantially reduces length bias without requiring additional forward passes and while preserving meaningful within-item preferences among candidates.

## 4.5. Comparison to baselines

We now compare Bayesian accuracy to standard accuracy, byte-normalized accuracy, PMI, and ANPMI in terms of their overall length bias. For each model and scoring rule $S$, we summarize length bias by the mean absolute correlation

$$\overline{|\tau|} \equiv \frac{1}{|\mathcal{B}|} \sum_{\mathcal{D} \in \mathcal{B}} \big|\tau(S, \mathcal{D})\big|, \qquad (25)$$

*Table 3.* Length bias under Bayesian accuracy.

| Benchmark | ARC | ARC German | HellaSwag | MMLU Cloze | MMLU Full-Text | OpenbookQA | SciQ | WinoGrande |
|---|---|---|---|---|---|---|---|---|
| Llama 3.2 1B | 0.06 | 0.04 | -0.02 | 0.06 | -0.02 | 0.05 | 0.07 | 0.01 |
| Llama 3.2 3B | 0.04 | 0.05 | -0.03 | 0.05 | 0.07 | 0.05 | 0.04 | 0.05 |
| Llama 3.1 8B | 0.03 | 0.03 | -0.04 | 0.06 | 0.02 | 0.06 | 0.03 | -0.01 |
| Llama 3.1 70B | 0.02 | 0.03 | -0.05 | 0.07 | 0.03 | 0.03 | 0.03 | 0.00 |
| Qwen3 600M | 0.06 | 0.07 | -0.01 | 0.06 | 0.00 | 0.06 | 0.07 | 0.02 |
| Qwen3 1.7B | 0.05 | 0.04 | -0.02 | 0.05 | -0.04 | 0.05 | 0.06 | 0.05 |
| Qwen3 4B | 0.03 | 0.05 | -0.03 | 0.08 | -0.02 | 0.04 | 0.06 | 0.01 |
| Qwen3 8B | 0.04 | 0.04 | -0.03 | 0.05 | 0.00 | 0.02 | 0.05 | 0.01 |
| Qwen3 14B | 0.03 | 0.01 | -0.04 | 0.07 | 0.02 | 0.04 | 0.05 | 0.04 |
| Mistral 7B | 0.02 | 0.05 | -0.04 | 0.06 | 0.03 | 0.03 | 0.03 | -0.01 |
| Pythia 410M | 0.07 | 0.03 | 0.00 | 0.01 | -0.16 | 0.08 | 0.10 | 0.06 |
| SmolLM 135M | 0.07 | 0.04 | -0.01 | 0.05 | -0.13 | 0.08 | 0.08 | 0.07 |
| Qwen3 600M Instruct | 0.06 | 0.07 | -0.02 | 0.04 | 0.03 | 0.06 | 0.09 | 0.00 |
| Qwen3 1.7B Instruct | 0.06 | 0.02 | -0.07 | 0.08 | 0.03 | 0.08 | 0.10 | 0.01 |
| Qwen3 4B Instruct | 0.05 | 0.05 | -0.07 | 0.04 | 0.00 | 0.11 | 0.09 | 0.02 |
| Qwen3 8B Instruct | 0.02 | 0.02 | -0.07 | -0.01 | 0.03 | 0.07 | 0.05 | -0.01 |
| Qwen3 14B Instruct | 0.03 | 0.02 | -0.08 | 0.05 | 0.04 | 0.08 | 0.05 | 0.05 |

averaged over the benchmarks $\mathcal{B}$ from Section 3.3. Table 4 reports $\overline{|\tau|}$ per model and metric (lower is better).

*Table 4.* Average absolute length bias $\overline{|\tau|}$ per model and scoring rule, aggregated over all benchmarks. *Standard* and *Norm* correspond to unnormalized and byte-normalized accuracy, respectively. Lower is better.

| Model | Standard | Norm | PMI | ANPMI | Bayes |
|---|---|---|---|---|---|
| Llama 3.2 1B | 0.24 | 0.32 | 0.16 | 0.11 | **0.04** |
| Llama 3.2 3B | 0.22 | 0.29 | 0.18 | 0.12 | **0.05** |
| Llama 3.1 8B | 0.20 | 0.27 | 0.17 | 0.11 | **0.03** |
| Llama 3.1 70B | 0.18 | 0.26 | 0.15 | 0.07 | **0.03** |
| Qwen3 600M | 0.22 | 0.31 | 0.18 | 0.11 | **0.04** |
| Qwen3 1.7B | 0.21 | 0.27 | 0.17 | 0.09 | **0.05** |
| Qwen3 4B | 0.20 | 0.26 | 0.15 | 0.10 | **0.04** |
| Qwen3 8B | 0.20 | 0.24 | 0.15 | 0.09 | **0.03** |
| Qwen3 14B | 0.19 | 0.25 | 0.16 | 0.10 | **0.04** |
| Mistral 7B | 0.21 | 0.27 | 0.17 | 0.10 | **0.03** |
| Pythia 410M | 0.31 | 0.32 | 0.17 | 0.11 | **0.06** |
| SmolLM 135M | 0.29 | 0.30 | 0.14 | 0.13 | **0.07** |
| Qwen3 600M Instruct | 0.22 | 0.40 | 0.11 | 0.07 | **0.05** |
| Qwen3 1.7B Instruct | 0.20 | 0.41 | 0.11 | **0.06** | **0.06** |
| Qwen3 4B Instruct | 0.19 | 0.42 | 0.10 | **0.05** | 0.06 |
| Qwen3 8B Instruct | 0.17 | 0.38 | 0.07 | 0.06 | **0.04** |
| Qwen3 14B Instruct | 0.17 | 0.42 | 0.08 | 0.06 | **0.05** |

Byte-normalized accuracy (*Norm*) exhibits the largest mean absolute length bias, often substantially worse than standard accuracy. PMI and ANPMI reduce bias relative to both, and ANPMI consistently improves on PMI, but both require unconditional log-likelihoods for every candidate, doubling evaluation cost and still leaving non-trivial residual bias. Bayesian accuracy attains the smallest or near-smallest $\overline{|\tau|}$ for all models: values in the range $0.03$–$0.07$ correspond to roughly a $4\times$–$8\times$ reduction in length bias relative to normalized accuracy and typically a several-fold improvement over PMI and ANPMI, for both base and instruction-tuned models. We therefore recommend Bayesian accuracy as a simple length-corrected default, reserving standard accuracy for regimes where length is provably irrelevant and using normalized accuracy only when its low bias has been empirically verified.

## 5. Conclusion

Multiple-choice benchmarks that score candidates by summed conditional log-likelihood are convenient and widely used, but this additivity induces systematic length bias: across models and datasets we find that total log-likelihoods scale approximately linearly with completion length, so standard accuracy tends to prefer shorter answers, while naïve length-normalization often over-corrects and favors longer ones instead. We quantify these effects via Kendall's $\tau$ between candidate lengths and scores within each example, which highlights when length can be ignored (locally determined answers or near-equal-length candidates) and when both standard and normalized accuracy become unreliable.

Motivated by the empirical score–length relationship, we introduce Bayesian accuracy: a drop-in replacement for standard accuracy that incorporates an explicit exponential prior over completion lengths and subtracts a learned global length trend. Bayesian accuracy only uses conditional log-likelihoods, requires no additional forward passes, and consistently reduces measured length bias by a large factor compared to standard, normalized, PMI, and ANPMI-based scoring across benchmarks and models. This addresses a measurable nuisance effect in likelihood-based evaluation, while broader questions of evaluation faithfulness, such as agreement with downstream capabilities, remain important directions for future work. We therefore view Bayesian accuracy as a simple, robust default for likelihood-based multiple-choice evaluation.

## Acknowledgements

I thank Tom Burns for many helpful brainstorming sessions and early discussions that shaped the ideas in this paper.

## Impact Statement

This work proposes a simple correction for length bias in likelihood-based multiple-choice evaluation of language models. More reliable scoring rules can change comparative model rankings and encourage fairer, more robust benchmark design, but they do not address other sources of bias or harmful behavior in the underlying models, which must be studied separately.

## Financial Conflict of Interest

The author declares no financial or other substantive conflicts of interest that could reasonably be perceived to influence this work.

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

# A. Appendix

## A.1. Details of Estimating the Length Decay Factor

A naïve way to estimate the global slope $b$ would be to pool all candidates and regress $\ell_m^{(k)}$ directly on $n_m^{(k)}$ using ordinary least squares. This ignores the grouping by prompts. If prompts with larger offsets $\alpha_k$ systematically induce longer or shorter candidates, the pooled slope absorbs both the true length effect and prompt-level correlations:

$$\mathbb{E}[\hat{b}_{\mathrm{OLS}}] = b + \frac{\mathbb{E}[\mathrm{Cov}(\alpha, n)]}{\mathrm{Var}(n)}. \tag{26}$$

To avoid this, we estimate $b$ using only within-example differences, which cancel the prompt-specific intercepts. For each example $k$ and candidate pair $(i, j)$, define

$$\Delta\ell_{ij}^{(k)} = \ell_\theta(c_i^{(k)} \mid x^{(k)}) - \ell_\theta(c_j^{(k)} \mid x^{(k)}), \tag{27}$$

$$\Delta n_{ij}^{(k)} = n_i^{(k)} - n_j^{(k)}. \tag{28}$$

Under the linear model in Equation 17, these differences satisfy

$$\Delta\ell_{ij}^{(k)} = \beta_k \, \Delta n_{ij}^{(k)} + \varepsilon_{ij}^{(k)}, \tag{29}$$

where $\varepsilon_{ij}^{(k)} \equiv \varepsilon_i^{(k)} - \varepsilon_j^{(k)}$ is zero-mean. Treating the $\beta_k$ as noisy draws around a common mean $b$, the corresponding pairwise least-squares estimator is

$$\hat{b} = \frac{\sum_k \sum_{i,j} \Delta\ell_{ij}^{(k)} \, \Delta n_{ij}^{(k)}}{\sum_k \sum_{i,j} (\Delta n_{ij}^{(k)})^2}. \tag{30}$$

The symmetry $\Delta n_{ij}^{(k)} = -\Delta n_{ji}^{(k)}$ implies that these differences are automatically mean-centered. The next subsection rewrites the same estimator in the computationally cheaper centered form used in Equation 24.

## A.2. Mean-Centered Form of the Length Decay Estimator

The pairwise estimator in (30) is naturally written as a double sum over all candidate pairs, which scales as $O(M_k^2)$ in the number of candidates $M_k$ for each prompt. In our setting $M_k$ is typically small, so the quadratic scaling is not a fundamental bottleneck, but for larger candidate sets it is helpful to avoid quadratic dependence. In this subsection we show that such pairwise sums can be rewritten exactly as single sums over mean-centered quantities, reducing the per-prompt computational cost from $O(M_k^2)$ to $O(M_k)$ while leaving the value of the estimator unchanged.

Let $x_1, \ldots, x_n$ and $y_1, \ldots, y_n$ be real numbers, and define the means

$$\bar{x} = \frac{1}{n} \sum_{i=1}^n x_i, \qquad \bar{y} = \frac{1}{n} \sum_{i=1}^n y_i. \tag{31}$$

Consider

$$S \equiv \sum_{i,j=1}^n (x_i - x_j)(y_i - y_j). \tag{32}$$

Expanding the product inside the sum gives

$$S = \sum_{i,j} (x_i y_i - x_i y_j - x_j y_i + x_j y_j) \tag{33}$$

$$= 2n \sum_i x_i y_i - 2 \Big(\sum_i x_i\Big)\Big(\sum_i y_i\Big) \tag{34}$$

$$= 2n \Big(\sum_i x_i y_i - n \bar{x} \, \bar{y}\Big). \tag{35}$$

Finally, note that

$$\sum_{i=1}^{n}(x_i - \bar{x})(y_i - \bar{y}) = \sum_{i} x_i y_i - n\bar{x}\,\bar{y},$$

so we arrive at the identity

$$\sum_{i,j=1}^{n}(x_i - x_j)(y_i - y_j) = 2n\sum_{i=1}^{n}(x_i - \bar{x})(y_i - \bar{y}). \tag{36}$$

Thus, a quadratic double sum over all pairs $(i,j)$ can be replaced by a linear single sum over mean-centered terms $(x_i - \bar{x})(y_i - \bar{y})$. Substituting these expressions into the definition of $\hat{b}$ and cancelling the common factor 2 yields the equivalent mean-centered form used in the main text

$$\hat{b} = \frac{\sum_{k} M_k \sum_{i=1}^{M_k}\left(n_i^{(k)} - \bar{n}^{(k)}\right)\left(\ell_i^{(k)} - \bar{\ell}^{(k)}\right)}{\sum_{k} M_k \sum_{i=1}^{M_k}\left(n_i^{(k)} - \bar{n}^{(k)}\right)^2}. \tag{37}$$

### A.3. Algorithm

---

**Algorithm 1** Estimate global length decay $b$ via within-prompt centering

---

**Input:** For each question $k$, candidate lengths $n_i^{(k)}$ and log-likelihoods $\ell_i^{(k)}$
**Output:** Estimated length decay factor $\hat{b}$

$num \leftarrow 0$
$den \leftarrow 0$
**for** each question $k$ **do**
    $M \leftarrow$ number of candidates for question $k$
    **if** $M \leq 1$ **or** $n_1^{(k)} = n_2^{(k)} = \cdots = n_M^{(k)}$ **then**
        **continue** {No within-question length variation}
    **end if**
    $\bar{n} \leftarrow \frac{1}{M}\sum_{i=1}^{M} n_i^{(k)}$
    $\bar{\ell} \leftarrow \frac{1}{M}\sum_{i=1}^{M} \ell_i^{(k)}$
    $cov \leftarrow M\sum_{i=1}^{M}\left(n_i^{(k)} - \bar{n}\right)\left(\ell_i^{(k)} - \bar{\ell}\right)$
    $var \leftarrow M\sum_{i=1}^{M}\left(n_i^{(k)} - \bar{n}\right)^2$
    $num \leftarrow num + cov$
    $den \leftarrow den + var$
**end for**
**if** $den = 0$ **then**
    $\hat{b} \leftarrow 0$ {No length variation across any question}
**else**
    $\hat{b} \leftarrow num/den$
**end if**
**return** $\hat{b}$

---

### A.4. Benchmark Details

**ARC.** The AI2 Reasoning Challenge (ARC) is a multiple-choice question answering dataset of 7,787 science questions from U.S. grade-school exams, split into an "Easy" and a more difficult "Challenge" subset. It is designed to require non-trivial scientific and commonsense reasoning rather than simple pattern matching or retrieval.

**HellaSwag.** HellaSwag is an adversarial commonsense benchmark in which models must choose the most plausible continuation of a short narrative or instructional context from four options. The roughly 70,000 examples are built from video captions and how-to articles, with distractor endings crafted to look fluent yet be semantically implausible to humans.

**MMLU.** The Massive Multitask Language Understanding (MMLU) benchmark aggregates 15,908 four-way multiple-choice questions across 57 academic subjects, spanning humanities, social and natural sciences, and professional domains. It is widely used as a proxy for broad world knowledge and domain-specific reasoning in language models.

**MMLU (full-text answer).** In our *MMLU Full-Text* variant, the underlying questions and answer options are unchanged, but the model is asked to output the full textual answer (e.g., "Paris") rather than the option label ("C"). Accuracy is computed by exact-match comparison between the generated answer and the corresponding gold option string.

**MMLU (cloze).** In the *MMLU cloze* variant, the multiple-choice options are removed from the prompt and the problem is phrased as a short-answer or fill-in-the-blank question. The model must produce the answer directly; at evaluation time we map the completion back onto the original MMLU answer key (after minor normalization) to determine correctness.

**SciQ.** SciQ is a four-option multiple-choice benchmark of about 14,000 general science questions modeled on elementary and middle-school exams. Questions often come with associated source passages, and the dataset is commonly used as a mid-difficulty science QA task between simple fact recall and more demanding benchmarks such as ARC.

**WinoGrande.** WinoGrande is a large-scale pronoun resolution benchmark inspired by the Winograd Schema Challenge. Each example is a short sentence with an ambiguous pronoun and two candidate antecedents; the model must decide which option makes the sentence coherent, probing fine-grained commonsense reasoning while controlling for dataset artifacts.

### A.5. Benchmark Prompt Examples

The following synthetic examples illustrate the prompt formats used for each benchmark in our evaluation. They are not taken from the original test sets.

- The quotation marks are not part of the text, they are there to highlight any potential white spacing

- Templating is highlighted in green

#### A.5.1. ARC

**User:** "Question: Which is an inherited characteristic on a horse?"
**Assistant:** "Answer:"
**Options:**

- " a long mane"
- " a steel shoe"
- " a missing tooth"
- " a bruised leg"

#### A.5.2. ARC GERMAN

**User:** "Frage: Welcher Prozess im Kohlenstoffkreislauf dauert am längsten?"
**Assistant:** "Antwort:"
**Options:**

- " Emission von Abfall"
- " Atmung bei Tieren"
- " Photosynthese bei Pflanzen"
- " Bildung von fossilen Brennstoffen"

A.5.3. HELLASWAG

**User:** "Cutting the grass: A man is kneeling down on grass. He"
**Options:**

- " uses a polishing brush on a shoe."
- " has a heavy work out machine in his arms."
- " is using a green brush to clean off the grass."
- " is clipping the grass with large scissors."

A.5.4. MMLU FULL-TEXT

**User:** "The following are multiple choice questions (with possible answers) about machine learning. Answer with the full text of the correct answer.

Question: Suppose your model is overfitting. Which of the following is NOT a valid way to try and reduce the overfitting?
Possible answers:
- Increase the amount of training data.
- Improve the optimisation algorithm being used for error minimisation.
- Decrease the model complexity.
- Reduce the noise in the training data."
**Assistant:** "Answer:"
**Options:**

- " Increase the amount of training data."
- " Improve the optimisation algorithm being used for error minimisation."
- " Decrease the model complexity."
- " Reduce the noise in the training data."

A.5.5. MMLU CLOZE

**User:** "The following are multiple choice questions (with possible answers) about machine learning. Answer with the full text of the correct answer.

Question: Suppose your model is overfitting. Which of the following is NOT a valid way to try and reduce the overfitting?"
**Assistant:** "Answer:"
**Options:**

- " Increase the amount of training data."
- " Improve the optimisation algorithm being used for error minimisation."
- " Decrease the model complexity."
- " Reduce the noise in the training data."

A.5.6. OPENBOOKQA

**User:** "Climate change has sped up dramatically because"
**Options:**

- " $CO_2$ production has accelerated"
- " a rapid decline the production of carbon dioxide"
- " oxygen levels have spiked"
- " Fe has been present in an overabundance"

## A.5.7. SCIQ

**User:** "Question: Anemia is a disease that affects what?"
**Assistant:** "Answer:"
**Options:**

- " Brain"
- " Heart"
- " Kidney"
- " Blood"

## A.5.8. WINOGRANDE

**User:** "Felicia ran out of shirts and borrowed one from Patricia, but"
**Options:**

- " Felicia didn't ask permission ahead of time."
- " Patricia didn't ask permission ahead of time."

## A.6. Few-shot Results

*Table 5.* Length bias for few-shot standard (unnormalized) accuracy. Entries are Kendall's $\tau$ between candidate length and score for each model–benchmark pair (negative values indicate a tendency to favor shorter completions).

| Benchmark | ARC | ARC German | HellaSwag | MMLU Cloze | MMLU Full-Text | OpenbookQA | SciQ | WinoGrande |
|---|---|---|---|---|---|---|---|---|
| Llama 3.2 1B | -0.19 | -0.28 | -0.55 | -0.33 | -0.07 | -0.30 | -0.06 | 0.03 |
| Llama 3.2 3B | -0.17 | -0.25 | -0.51 | -0.36 | 0.08 | -0.27 | -0.05 | 0.00 |
| Llama 3.1 8B | -0.15 | -0.24 | -0.49 | -0.30 | 0.06 | -0.25 | -0.02 | -0.03 |
| Llama 3.1 70B | -0.13 | -0.17 | -0.45 | -0.25 | NaN | -0.23 | -0.03 | -0.02 |
| Qwen3 600M | -0.17 | -0.25 | -0.58 | -0.30 | 0.10 | -0.30 | -0.05 | 0.03 |
| Qwen3 1.7B | -0.15 | -0.24 | -0.54 | -0.31 | 0.06 | -0.28 | -0.02 | 0.07 |
| Qwen3 4B | -0.14 | -0.21 | -0.52 | -0.29 | 0.03 | -0.25 | -0.02 | -0.02 |
| Qwen3 8B | -0.13 | -0.19 | -0.50 | -0.26 | 0.06 | -0.26 | -0.02 | 0.02 |
| Qwen3 14B | -0.12 | -0.18 | -0.48 | -0.27 | 0.03 | -0.24 | -0.03 | 0.00 |
| Mistral 7B | -0.14 | -0.22 | -0.48 | -0.26 | 0.09 | -0.25 | -0.04 | -0.01 |
| Pythia 410M | -0.25 | -0.39 | -0.61 | -0.45 | -0.15 | -0.33 | -0.14 | -0.04 |
| SmolLM 135M | -0.22 | -0.43 | -0.60 | -0.40 | -0.09 | -0.30 | -0.08 | 0.00 |
| Qwen3 600M Instruct | -0.15 | -0.26 | -0.57 | -0.32 | 0.09 | -0.27 | -0.05 | 0.09 |
| Qwen3 1.7B Instruct | -0.12 | -0.20 | -0.53 | -0.25 | 0.08 | -0.21 | -0.03 | 0.09 |
| Qwen3 4B Instruct | -0.11 | -0.17 | -0.53 | -0.21 | 0.08 | -0.20 | -0.02 | 0.15 |
| Qwen3 8B Instruct | -0.08 | -0.13 | -0.51 | -0.20 | 0.07 | -0.20 | -0.02 | -0.01 |
| Qwen3 14B Instruct | -0.10 | -0.14 | -0.51 | -0.20 | 0.06 | -0.20 | -0.03 | 0.03 |

*Table 6.* Length bias for few-shot byte-normalized accuracy. Entries are Kendall's $\tau$ between candidate length and score; positive values indicate a tendency to favor longer completions.

| Benchmark | ARC | ARC German | HellaSwag | MMLU Cloze | MMLU Full-Text | OpenbookQA | SciQ | WinoGrande |
|---|---|---|---|---|---|---|---|---|
| Llama 3.2 1B | 0.20 | 0.26 | 0.03 | 0.22 | 0.37 | 0.30 | 0.20 | 0.47 |
| Llama 3.2 3B | 0.19 | 0.25 | 0.03 | 0.18 | 0.39 | 0.30 | 0.22 | 0.40 |
| Llama 3.1 8B | 0.18 | 0.25 | 0.03 | 0.19 | 0.38 | 0.32 | 0.22 | 0.36 |
| Llama 3.1 70B | 0.20 | 0.25 | 0.03 | 0.21 | NaN | 0.32 | 0.22 | 0.28 |
| Qwen3 600M | 0.17 | 0.25 | 0.04 | 0.20 | 0.36 | 0.28 | 0.18 | 0.50 |
| Qwen3 1.7B | 0.17 | 0.26 | 0.04 | 0.17 | 0.33 | 0.29 | 0.21 | 0.42 |
| Qwen3 4B | 0.18 | 0.25 | 0.03 | 0.17 | 0.29 | 0.30 | 0.21 | 0.36 |
| Qwen3 8B | 0.19 | 0.25 | 0.03 | 0.17 | 0.29 | 0.29 | 0.21 | 0.36 |
| Qwen3 14B | 0.19 | 0.25 | 0.03 | 0.17 | 0.29 | 0.29 | 0.22 | 0.31 |
| Mistral 7B | 0.16 | 0.23 | 0.03 | 0.19 | 0.32 | 0.32 | 0.21 | 0.33 |
| Pythia 410M | 0.22 | 0.27 | 0.05 | 0.14 | 0.18 | 0.29 | 0.26 | 0.47 |
| SmolLM 135M | 0.18 | 0.24 | 0.04 | 0.18 | 0.15 | 0.31 | 0.22 | 0.45 |
| Qwen3 600M Instruct | 0.18 | 0.25 | 0.24 | 0.17 | 0.24 | 0.52 | 0.20 | 0.46 |
| Qwen3 1.7B Instruct | 0.17 | 0.21 | 0.21 | 0.15 | 0.26 | 0.62 | 0.23 | 0.39 |
| Qwen3 4B Instruct | 0.21 | 0.24 | 0.24 | 0.24 | 0.28 | 0.53 | 0.26 | 0.41 |
| Qwen3 8B Instruct | 0.21 | 0.25 | 0.30 | 0.25 | 0.28 | 0.52 | 0.27 | 0.25 |
| Qwen3 14B Instruct | 0.25 | 0.28 | 0.33 | 0.27 | 0.28 | 0.55 | 0.28 | 0.27 |

*Table 7.* Length bias for few-shot Bayesian accuracy. Entries are Kendall's $\tau$ between candidate length and Bayesian-corrected score.

| Benchmark | ARC | ARC German | HellaSwag | MMLU Cloze | MMLU Full-Text | OpenbookQA | SciQ | WinoGrande |
|---|---|---|---|---|---|---|---|---|
| Llama 3.2 1B | 0.04 | 0.05 | -0.02 | 0.14 | 0.01 | 0.05 | 0.06 | 0.12 |
| Llama 3.2 3B | 0.03 | 0.06 | -0.04 | 0.14 | 0.10 | 0.04 | 0.04 | 0.04 |
| Llama 3.1 8B | 0.03 | 0.04 | -0.05 | 0.13 | 0.05 | 0.05 | 0.04 | 0.00 |
| Llama 3.1 70B | 0.02 | 0.03 | -0.05 | 0.13 | NaN | 0.05 | 0.04 | 0.00 |
| Qwen3 600M | 0.03 | 0.07 | -0.02 | 0.16 | 0.07 | 0.06 | 0.05 | 0.05 |
| Qwen3 1.7B | 0.03 | 0.06 | -0.02 | 0.15 | 0.04 | 0.05 | 0.06 | 0.11 |
| Qwen3 4B | 0.03 | 0.05 | -0.03 | 0.13 | -0.02 | 0.06 | 0.06 | 0.03 |
| Qwen3 8B | 0.03 | 0.03 | -0.04 | 0.11 | 0.02 | 0.03 | 0.05 | 0.06 |
| Qwen3 14B | 0.04 | 0.04 | -0.04 | 0.11 | 0.01 | 0.03 | 0.06 | 0.04 |
| Mistral 7B | 0.02 | 0.05 | -0.05 | 0.14 | 0.09 | 0.06 | 0.04 | 0.07 |
| Pythia 410M | 0.06 | 0.04 | -0.01 | 0.11 | -0.09 | 0.09 | 0.10 | 0.04 |
| SmolLM 135M | 0.05 | 0.04 | -0.01 | 0.15 | -0.06 | 0.07 | 0.06 | 0.11 |
| Qwen3 600M Instruct | 0.03 | 0.06 | -0.05 | 0.11 | 0.10 | 0.04 | 0.06 | 0.09 |
| Qwen3 1.7B Instruct | 0.03 | 0.03 | -0.19 | 0.09 | 0.08 | 0.09 | 0.07 | 0.03 |
| Qwen3 4B Instruct | 0.03 | 0.01 | -0.19 | 0.07 | 0.05 | 0.06 | 0.06 | 0.09 |
| Qwen3 8B Instruct | 0.00 | 0.01 | -0.19 | 0.00 | 0.05 | -0.05 | 0.04 | -0.02 |
| Qwen3 14B Instruct | -0.01 | -0.01 | -0.19 | 0.04 | 0.03 | -0.01 | 0.04 | 0.02 |

## A.7. (AN)PMI Results

*Table 8.* Length bias for zero-shot ANPMI scoring. Entries are Kendall's $\tau$ between candidate length and ANPMI score.

| Benchmark | ARC | ARC German | HellaSwag | MMLU Cloze | MMLU Full-Text | OpenbookQA | SciQ | WinoGrande |
|---|---|---|---|---|---|---|---|---|
| Llama 3.2 1B | -0.04 | -0.08 | -0.28 | -0.18 | 0.23 | -0.06 | 0.01 | -0.02 |
| Llama 3.2 3B | -0.05 | -0.10 | -0.27 | -0.14 | 0.29 | -0.06 | 0.02 | 0.02 |
| Llama 3.1 8B | -0.04 | -0.09 | -0.24 | -0.09 | 0.26 | -0.04 | 0.03 | -0.07 |
| Llama 3.1 70B | -0.03 | -0.04 | -0.22 | -0.02 | 0.17 | -0.03 | 0.04 | -0.03 |
| Qwen3 600M | -0.02 | -0.07 | -0.30 | -0.15 | 0.22 | -0.04 | 0.03 | 0.05 |
| Qwen3 1.7B | -0.03 | -0.07 | -0.27 | -0.16 | 0.13 | -0.01 | 0.03 | 0.04 |
| Qwen3 4B | -0.05 | -0.08 | -0.27 | -0.15 | 0.16 | -0.04 | 0.04 | -0.05 |
| Qwen3 8B | -0.05 | -0.08 | -0.26 | -0.14 | 0.14 | -0.03 | 0.04 | 0.00 |
| Qwen3 14B | -0.05 | -0.08 | -0.24 | -0.13 | 0.18 | -0.02 | 0.04 | 0.02 |
| Mistral 7B | -0.04 | -0.06 | -0.23 | -0.14 | 0.24 | -0.02 | 0.03 | -0.01 |
| Pythia 410M | -0.06 | -0.11 | -0.31 | -0.14 | 0.10 | -0.12 | 0.00 | 0.00 |
| SmolLM 135M | -0.06 | -0.20 | -0.30 | -0.18 | 0.15 | -0.08 | -0.01 | 0.02 |
| Qwen3 600M Instruct | -0.01 | -0.05 | -0.07 | -0.18 | 0.12 | 0.06 | 0.04 | 0.00 |
| Qwen3 1.7B Instruct | 0.03 | -0.01 | -0.01 | 0.01 | 0.16 | 0.14 | 0.08 | 0.04 |
| Qwen3 4B Instruct | 0.00 | -0.03 | -0.03 | -0.01 | 0.15 | 0.07 | 0.05 | 0.09 |
| Qwen3 8B Instruct | -0.02 | -0.02 | -0.05 | -0.07 | 0.15 | 0.12 | 0.02 | 0.01 |
| Qwen3 14B Instruct | -0.02 | -0.05 | -0.11 | -0.02 | 0.15 | 0.10 | 0.04 | 0.01 |

*Table 9.* Length bias for zero-shot PMI scoring. Entries are Kendall's $\tau$ between candidate length and PMI score.

| Benchmark | ARC | ARC German | HellaSwag | MMLU Cloze | MMLU Full-Text | OpenbookQA | SciQ | WinoGrande |
|---|---|---|---|---|---|---|---|---|
| Llama 3.2 1B | 0.10 | 0.15 | 0.13 | 0.19 | 0.45 | 0.11 | 0.11 | 0.07 |
| Llama 3.2 3B | 0.11 | 0.14 | 0.12 | 0.22 | 0.49 | 0.16 | 0.10 | 0.06 |
| Llama 3.1 8B | 0.10 | 0.13 | 0.11 | 0.24 | 0.44 | 0.15 | 0.09 | -0.07 |
| Llama 3.1 70B | 0.09 | 0.14 | 0.10 | 0.22 | 0.40 | 0.13 | 0.10 | -0.01 |
| Qwen3 600M | 0.13 | 0.13 | 0.14 | 0.17 | 0.45 | 0.14 | 0.13 | 0.11 |
| Qwen3 1.7B | 0.12 | 0.13 | 0.12 | 0.17 | 0.45 | 0.14 | 0.12 | 0.09 |
| Qwen3 4B | 0.09 | 0.14 | 0.11 | 0.18 | 0.40 | 0.13 | 0.11 | 0.02 |
| Qwen3 8B | 0.09 | 0.14 | 0.11 | 0.18 | 0.40 | 0.15 | 0.13 | 0.04 |
| Qwen3 14B | 0.07 | 0.13 | 0.09 | 0.24 | 0.42 | 0.14 | 0.12 | 0.06 |
| Mistral 7B | 0.11 | 0.16 | 0.12 | 0.20 | 0.48 | 0.14 | 0.12 | 0.05 |
| Pythia 410M | 0.09 | 0.15 | 0.17 | 0.20 | 0.46 | 0.07 | 0.14 | 0.08 |
| SmolLM 135M | 0.08 | 0.11 | 0.15 | 0.15 | 0.42 | 0.06 | 0.10 | 0.09 |
| Qwen3 600M Instruct | 0.08 | 0.11 | 0.10 | 0.06 | 0.39 | 0.05 | 0.08 | 0.01 |
| Qwen3 1.7B Instruct | 0.04 | 0.05 | 0.12 | 0.14 | 0.31 | 0.03 | 0.09 | 0.08 |
| Qwen3 4B Instruct | 0.02 | 0.06 | 0.12 | 0.08 | 0.28 | 0.05 | 0.08 | 0.07 |
| Qwen3 8B Instruct | 0.01 | 0.06 | 0.04 | 0.04 | 0.29 | 0.07 | 0.06 | 0.01 |
| Qwen3 14B Instruct | 0.01 | 0.05 | 0.06 | 0.05 | 0.29 | 0.08 | 0.06 | 0.01 |

## A.8. Variance Scaling

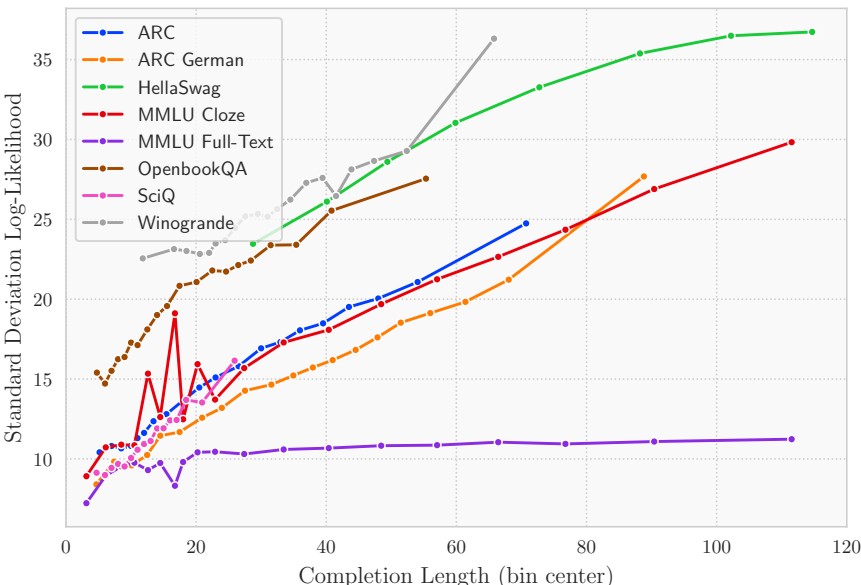

*Figure 3.* Standard deviation of conditional log-likelihood $\ell_\theta(c \mid x)/n_{\text{byte}}$ versus completion byte length, aggregated over all models. For readability, completion lengths are restricted to a maximum of 120 bytes and results are binned.

## A.9. Token-Level Length Bias Results

*Table 10.* Average absolute length bias $\overline{|\tau|}$ per model and scoring rule when length is measured in tokens, aggregated over all benchmarks. *Standard* is unnormalized accuracy, while *Norm* and *Bayes* use token length for normalization and length correction, respectively. Lower is better.

| Model | Standard | Norm | PMI | ANPMI | Bayes |
|---|---|---|---|---|---|
| Llama 3.2 1B | 0.29 | 0.41 | 0.31 | 0.17 | **0.05** |
| Llama 3.2 3B | 0.26 | 0.38 | 0.30 | 0.15 | **0.06** |
| Llama 3.1 8B | 0.23 | 0.35 | 0.29 | 0.14 | **0.04** |
| Llama 3.1 70B | 0.22 | 0.33 | 0.26 | 0.10 | **0.04** |
| Qwen3 600M | 0.28 | 0.39 | 0.25 | 0.15 | **0.06** |
| Qwen3 1.7B | 0.25 | 0.35 | 0.26 | 0.13 | **0.05** |
| Qwen3 4B | 0.24 | 0.33 | 0.23 | 0.12 | **0.05** |
| Qwen3 8B | 0.22 | 0.33 | 0.23 | 0.12 | **0.03** |
| Qwen3 14B | 0.23 | 0.32 | 0.25 | 0.12 | **0.05** |
| Mistral 7B | 0.22 | 0.36 | 0.26 | 0.12 | **0.05** |
| Pythia 410M | 0.37 | 0.40 | 0.25 | 0.14 | **0.06** |
| SmolLM 135M | 0.34 | 0.37 | 0.23 | 0.17 | **0.05** |
| Qwen3 600M Instruct | 0.29 | 0.48 | 0.20 | 0.13 | **0.05** |
| Qwen3 1.7B Instruct | 0.26 | 0.46 | 0.18 | 0.14 | **0.06** |
| Qwen3 4B Instruct | 0.26 | 0.47 | 0.15 | 0.09 | **0.04** |
| Qwen3 8B Instruct | 0.23 | 0.45 | 0.15 | 0.12 | **0.03** |
| Qwen3 14B Instruct | 0.23 | 0.48 | 0.15 | 0.11 | **0.03** |

