# OpenReview forum: "Accuracy and Normalized Accuracy under Length Bias: Analysis, Guidelines, and a Bayesian Alternative"
_ICML.cc/2026/Conference — ICML 2026 regular_

### Official Review · Reviewer_GJzB · 2026-03-11

**Soundness:** 3
**Presentation:** 3
**Significance:** 2
**Originality:** 3
**Overall Recommendation:** 4
**Confidence:** 4

**Summary:**

This paper studies length bias in likelihood-based multiple-choice evaluation for LLMs. The authors argue that standard scoring by summed conditional log-likelihood systematically favors shorter answers, while naive length normalization often over-corrects and favors longer ones. To make this precise, the paper defines length bias as the average within-example Kendall rank correlation between candidate length and candidate score. Motivated by an empirical approximately linear score-length relationship, the paper proposes Bayesian accuracy, which introduces an explicit prior over completion length and yields a corrected score of the form is estimated using within-example differences to avoid prompt-level intercept confounding. Experiments on ARC, ARC German, HellaSwag, MMLU (Cloze and Full-text), OpenbookQA, SciQ, and Winogrande, across LLaMA, Qwen, Mistral, Pythia, and SmolLM families, show that Bayesian accuracy substantially reduces measured length bias relative to standard accuracy, byte-normalized accuracy, PMI, and ANPMI, while requiring no extra forward passes

**Compliance With Llm Reviewing Policy:**

Affirmed.

**Ethical Review Flag:**

Flag this paper for an ethics review.

**Key Questions For Authors:**

1. he paper shows that Bayesian accuracy reduces  ∣τ∣, but can the authors also show that it improves the quality of evaluation outcomes, for example by yielding more reliable model rankings or by better matching task correctness across benchmark formats?

2. How exactly is 𝑏  estimated in the few-shot setting? Is it re-estimated per prompt template or per exemplar set, and how sensitive are the results to this choice? The answer matters if the method is intended as a default evaluation rule rather than a retrospective correction

3. The main experiments primarily use byte length for tokenizer independence. Do the qualitative conclusions remain equally strong under token-length-based correction, and are there any settings where byte-based and token-based Bayesian corrections diverge materially?

4. Since the generative model already allows prompt-specific slopes  , have the authors considered item-adaptive or prompt-adaptive variants of the correction, and if so, how do they compare to the single global 𝑏  design in terms of robustness and overfitting

**Strengths And Weaknesses:**

Strengths.

1. The paper addresses a real and under-discussed evaluation problem. The observation that summed log-likelihood can introduce a structural preference for short completions is important for many widely used benchmarks, and the paper gives a clean formalization of this issue via Kendall’s τ, which makes the phenomenon measurable rather than anecdotal

2. The proposed method is simple and elegant. The Bayesian correction is easy to implement, remains a drop-in replacement for existing likelihood-based pipelines, and does not require additional model evaluations

3. The empirical section is fairly broad. The paper covers several benchmark families, includes both base and instruction-tuned models, and offers a useful analysis of when standard accuracy may actually be acceptable


Weakness

1. My main concern is that the empirical objective is largely reduction of score-length correlation, rather than demonstrating that the new metric leads to more faithful benchmark conclusions. The paper convincingly shows that Bayesian accuracy drives ∣τ∣ closer to zero, and Table 4 is strong on that point, but the main text gives much less analysis of whether this translates into better agreement with ground-truth correctness, more trustworthy model rankings, or more stable conclusions across prompt formats. In other words, the paper strongly optimizes the proposed bias metric, but the link from lower ∣τ∣ to “better evaluation” could be argued more directly

2. the practical protocol around estimating 𝑏  could be clarified. As written, 𝑏  is estimated separately for each model-dataset pair and then used to rescore that same benchmark. This is reasonable as a diagnostic correction, but it raises questions about benchmark-specific fitting: how stable is 𝑏  across subsamples, prompt templates, or few-shot exemplars, and should it be estimated on held-out data for a truly deployment-style evaluation protocol? The paper claims improved behavior in few-shot settings, but the main text does not fully spell out these operational details

3. the scope is somewhat narrow. The experiments are centered on zero-shot multiple-choice benchmarks, primarily using byte length, and the impact statement itself notes that the method only addresses one source of evaluation bias rather than other forms of bias or harmful behavior in the models. This is a fair limitation, but it also means the contribution is best viewed as a targeted evaluation correction, not a general solution to benchmarking reliability.

---

> ### Author Rebuttal · Authors · 2026-03-30
>
> We thank the reviewer for the thoughtful and constructive feedback. We are glad the reviewer finds the problem important and the method simple and practical.
>
> We agree that the link between lower $|\tau|$ and better evaluation conclusions should be made more explicit. Our core position is that answer length is a nuisance variable in likelihood-based multiple-choice evaluation, so a desirable scoring rule should be invariant to it. In that sense, lower $|\tau|$ is an important property, but not a complete notion of evaluation quality. Defining a more global notion of "better evaluation'"" is difficult without introducing further assumptions, e.g., that rankings should be preserved across benchmarks or prompt formats. Ultimately, the strongest notion would be agreement with downstream capabilities, which is much elaborate to measure and outside the scope of this paper. We will revise the paper to make this scope clearer and to present Bayesian accuracy as removing a measurable nuisance effect, while leaving broader questions of evaluation faithfulness to future work.
>
> We agree that the few-shot protocol should be explained more clearly. In our setup, the only difference between zero-shot and few-shot is that the few-shot prompt is longer because it includes exemplars. We use one fixed exemplar set per prompt configuration, rather than averaging over many possible exemplar combinations. Accordingly, b is estimated separately for each model-dataset-prompt configuration, including the few-shot version with its longer prompt. We will make this protocol explicit in the revised text.
>
> Regarding the use of byte length, the method is agnostic to the choice of length unit: the derivation is unchanged if byte length is replaced by token length. We used bytes in the main text for tokenizer independence. We will add token-length results to the appendix and clarify that the qualitative conclusions remain the same, with only quantitative differences between byte- and token-based corrections.
>
> Finally, we chose a single global b per model-dataset-prompting configuration intentionally. Although the generative model allows prompt-specific slopes, item-adaptive corrections would need to estimate slopes from only a few candidates per question and would therefore be high-variance and prone to overfitting away genuine semantic differences. The global correction is a deliberate bias-variance tradeoff: it removes the dominant systematic linear trend while remaining stable, cheap, and easy to apply.

---

> > ### Author Rebuttal · Reviewer_GJzB · 2026-04-08
> >
> > The authors provide reasonable clarifications on all four questions. The scope of the contribution is now clearer, and the bias-variance justification for a global correction is sensible. I will keep
> >   my score.

---

### Official Review · Reviewer_NqHW · 2026-03-12

**Soundness:** 3
**Presentation:** 3
**Significance:** 3
**Originality:** 3
**Overall Recommendation:** 4
**Confidence:** 4

**Summary:**

This paper systematically analyzes the inherent length penalty bias in multiple-choice question evaluation based on log-likelihood, pointing out that traditional direct length normalization methods lead to over-correction towards longer texts.

**Compliance With Llm Reviewing Policy:**

Affirmed.

**Final Justification:**

Thank you to the author for the candid and clear clarification.

I agree that this method should be defined as a 'first-order heuristic correction for specific conditions.' Given that this method still has inherent theoretical limitations in handling the nonlinear dynamics of the first token and model distribution shifts (such as after fine-tuning), I believe Weak Accept is the most accurate assessment of its contributions and limitations, and I look forward to seeing the authors' promised supplementary discussion on limitations in the final version.

**Key Questions For Authors:**

Regarding the dynamic shift in vocabulary distribution that occurs during continuous learning, can this length prior be designed as an adaptive moving average update mechanism?

The prediction difficulty of generating the first token is often much higher than that of subsequent sequences. How can the probability decay of this non-linear start be accurately compensated within a linear correction framework?

After efficiently fine-tuning the parameters of a large model (e.g., LoRA), is it necessary to forcibly recalibrate the Bayesian prior to address changes in the model's tendency to generate different lengths?

**Limitations:**

The proposed solution is essentially only a post-processing heuristic at the scoring level and cannot truly eliminate the accumulated uncertainty of long sequence generation at the level of the model's internal feature representation.

**Strengths And Weaknesses:**

**Strengths**

A parameter-free Bayesian accuracy calculation method is proposed, which effectively neutralizes the linear correlation between length variables and the final score by introducing an explicit length prior.

Theoretical proof and empirical analysis are closely integrated, with a core focus on the systematic nature of fundamental evaluation operators in large language models.

The novel Bayesian scoring rule requires no additional forward propagation steps, offering good computational economy and broad applicability as a plug-and-play component.

**Weaknesses**

The core algorithm heavily relies on the prior assumption that "log-likelihood decays strictly linearly with length," which may no longer hold true in some extreme word segmenters (such as byte-based backoff) or specific language systems.

In scenarios with extremely small sample sizes (Few-shot), empirical estimates of the length prior probability are prone to drastic fluctuations, potentially leading to severe distortion in the final log probability correction.

This work, by merely addressing the penalty imposed by the physical length of the text at the numerical level, completely ignores the inherent preference of the model's training distribution for specific high-frequency words or syntactic structures.

In tasks such as multimodal visual question answering, the confidence of the answer is more influenced by the matching degree of visual features than simply the output length. Forcing linear correction in this case introduces significant new errors.

---

> ### Author Rebuttal · Authors · 2026-03-30
>
> We thank the reviewer for the thoughtful feedback and positive assessment. We agree that the contribution should be framed as a first-order correction for textual likelihood-based multiple-choice evaluation, rather than a universal debiasing method, and we will revise the paper accordingly.
>
> First, regarding the linearity assumption: we agree that strict linearity should not be claimed as a universal property of all tokenizers, languages, or model families. However, for the BPE tokenizers used by most contemporary autoregressive LLMs, the dominant empirical relationship between log-likelihood and completion length is approximately affine on the benchmarks we study. Bayesian accuracy is designed to remove this dominant first-order trend and indeed does not guarantee the removal of all possible nonlinear length effects. We will soften the wording accordingly and explicitly list unusual tokenizers/languages and nonlinear regimes as limitations.
>
> Second, regarding few-shot instability: in our few-shot experiments, “few-shot” refers to the prompting setup, not to estimating b from only a few examples. b is still estimated on the full evaluation split, typically hundreds to thousands of items. That said, we agree that with a genuinely small calibration set, b may become noisy. We will add this limitation and mention simple mitigations such as shrinkage toward b, held-out calibration, or pooling across related templates as possible solutions.
>
> Third, we fully agree that Bayesian accuracy does not remove all answer priors, such as preferences for frequent words or syntactic structures. This is by design: it targets a specific artifact induced by additive log-likelihood accumulation over length. We will make this scope clearer.
>
> Fourth, we agree that the current paper should not be interpreted as covering multimodal VQA or settings where confidence is dominated by visual grounding. Our theory and experiments are explicitly about textual multiple-choice evaluation based on conditional log-likelihoods.
>
> Regarding the reviewer’s questions:
> (1) Yes, an adaptive moving-average update of b under continual learning is a natural extension. This could be a future direction to explore.
> (2) The unusually difficult first token is captured primarily by the intercept term, and within-prompt differencing largely cancels it, though a piecewise or two-parameter extension could model short-length curvature more accurately.
> (3) Indeed, recalibration after LoRA/PEFT is likely needed as this changes the model's output distribution. However, this inexpensive since it uses the same conditional log-likelihoods that would already be used to reevaluate the model.

---

> > ### Author Rebuttal · Reviewer_NqHW · 2026-04-03
> >
> > Thank you to the author for the candid and clear clarification.
> >
> > I agree that this method should be defined as a 'first-order heuristic correction for specific conditions.' Given that this method still has inherent theoretical limitations in handling the nonlinear dynamics of the first token and model distribution shifts (such as after fine-tuning), I believe Weak Accept is the most accurate assessment of its contributions and limitations, and I look forward to seeing the authors' promised supplementary discussion on limitations in the final version.

---

### Official Review · Reviewer_WQmi · 2026-03-16

**Soundness:** 3
**Presentation:** 3
**Significance:** 3
**Originality:** 3
**Overall Recommendation:** 4
**Confidence:** 4

**Summary:**

This paper analyzes the length bias and scoring functions in multiple-choice benchmarks for LLMs. The authors show that standard scoring function based on summed conditional log-likelihood tends to favor shorter candidates, while length-normalized scoring tends to over-correct and induces the opposite bias toward longer candidates. To quantify this bias and trend, the paper defines length bias as the average Kendall rank correlation coefficient. Experiment results with a broad set of multiple-choice benchmarks and model families demonstrate that neither standard nor length-normalized scoring is reliable across settings.

Motivated by the observation that total conditional log-likelihood tends to scale approximately linearly with completion length, the paper proposes a simple “Bayesian accuracy” correction that subtracts a learned global linear length trend from each candidate score. This method requires no additional forward passes and substantially reduces measured length bias relative to standard accuracy, normalized accuracy, PMI, and ANPMI across the reported experiments.

**Compliance With Llm Reviewing Policy:**

Affirmed.

**Final Justification:**

Thank you for the clarifications. While certain limitations remain, I believe a weak accept remains appropriate, and I will maintain the score.

**Key Questions For Authors:**

1. The method estimates a global slope b using all examples and candidate pairs in a benchmark. How sensitive is this estimate to the size of the dataset? For example, does the method remain stable if b is estimated from a smaller subset of the data? Or, how reliable is the method when the dataset size is relatively small?
2. The experiments focus on zero-shot multiple-choice likelihood evaluation with fixed candidate completions. Do the authors expect the proposed correction to remain effective and applicable under other prompting setups, such as chain-of-thought prompting?

**Limitations:**

yes

**Strengths And Weaknesses:**

### Soundness:
The paper provides a careful empirical analysis showing that standard log-likelihood scoring in multiple-choice evaluation favors shorter answers, while length-normalized scoring often over-corrects and favors longer ones. The proposed Kendall-based metric for measuring length bias is well motivated, and the empirical study across multiple models and benchmarks supports the claims. The proposed Bayesian correction is simple, computationally cheap, and consistently reduces measured length bias. A potential limitation is that the slope parameter b is estimated using all examples in a benchmark, and the stability of this estimate under limited data or distribution shifts is not fully explored.

### Presentation:
The paper is generally clear and well organized, with a logical progression from empirical diagnosis to the proposed correction. One improvement would be to include a visualization analogous to Figures 1–2 showing the corrected score under the proposed method to directly confirm that the length trend is effectively reduced.

### Significance:
The work addresses a practical issue in widely used likelihood-based multiple-choice evaluation. Since many benchmarks rely on this evaluation protocol, understanding and correcting length bias can improve the reliability of model comparisons. However, the scope of the contribution is somewhat limited, as it focuses on a specific evaluation setting rather than a broader modeling problem. Nevertheless, the analysis is insightful and the proposed correction could be useful for practitioners employing likelihood-based scoring.

### Originality:
The paper’s originality mainly lies in its systematic analysis of length bias and the practical correction derived from this observation. The combination of empirical diagnosis, bias measurement, and a lightweight correction provides useful insights into evaluation practices.

---

> ### Author Rebuttal · Authors · 2026-03-30
>
> We thank the reviewer for the careful reading and constructive feedback. We are glad that the reviewer finds the empirical analysis, the Kendall-based bias measure, and the proposed correction useful. We will revise the paper to address the two main questions more explicitly and to improve the presentation.
>
> On the sensitivity of the estimate of b to dataset size: while the current draft does not include a dedicated subsampling experiment, the reported results already cover a fairly wide range of benchmark sizes, from 500 examples (OpenBookQA) to 14042 (MMLU). In particular, Bayesian accuracy remains effective on relatively small datasets such as OpenBookQA (500), SciQ (1000), ARC German (1172), and WinoGrande (1267), where it still substantially reduces length bias. We view this as evidence that the global slope estimate is reasonably stable in practice even when the dataset is not large. That said, we agree that an explicit subset-sensitivity analysis would strengthen the paper, and we will add this discussion in the revision.
>
> On applicability beyond the zero-shot setting: our few-shot results already suggest that the method is not specific to zero-shot prompting. More broadly, the correction applies whenever evaluation amounts to ranking a fixed candidate set by conditional log-likelihood. We have not yet evaluated chain-of-thought prompting, however, and do not want to overstate the claim. Chain-of-thought changes both the structure and the length distribution of completions, so it is plausible that similar effects arise, but this requires separate empirical study. We will add this as an explicit limitation and future direction.

---

> > ### Author Rebuttal · Reviewer_WQmi · 2026-04-03
> >
> > Thank you for the clarifications. While certain limitations remain, I believe a weak accept remains appropriate, and I will maintain the score.

---

### Decision · Program_Chairs · 2026-04-30

**Decision:**

Accept (regular)

**Comment:**

This paper analyzes length bias in likelihood-based multiple-choice evaluation and proposes a simple correction that removes the dominant linear length effect without extra computation.
All reviewers find the analysis clear and the method practical and technically sound.
The main concerns are about limited scope, reliance on approximate linearity, and whether reducing score-length correlation truly improves evaluation quality, which are resolved or clarified in the rebuttal discussions.